# Adapting and Explaining Time Series Foundation Models with Concepts

## Abstract

Foundation models for time series have achieved impressive performance on high-dimensional temporal data, but their opacity remains a barrier to adoption in domains where transparency and trust are essential. While Concept Activation Vectors (CAVs) have proven effective for aligning model predictions with human-understandable concepts in vision, their application to temporal data remains largely unexplored due to the complexity of sequential dynamics and the challenge of defining meaningful temporal concepts. In this work, we propose COFT, the first framework that adapts CAVs to foundation models for time series. COFT discovers dataset-specific temporal concepts through shapelet-based transformations and organizes them into a concept bank, enabling alignment between temporal subsequences and class labels. To integrate these concepts efficiently, COFT employs Low-Rank Adaptation (LoRA), allowing foundation models to internalize temporal concepts without sacrificing efficiency. Our experiments on diverse real-world benchmarks demonstrate that COFT not only improves interpretability quantified through concept alignment and CAV-based metrics, but also enhances predictive performance compared to state-of-the-art foundation models. Our work establishes a foundation for concept-based interpretability in time series modeling, bridging the gap between predictive power and transparent model reasoning.

## 1 Introduction

**Background**. Deep learning methods have become state-of-the-art for time series classification (Goswami et al., 2024; Ansari et al., 2024; Zhang et al., 2024) , with demonstrated impact in critical domains such as finance (Mai, 2024) and healthcare (Song et al., 2024; Pillai et al., 2025). Despite their high accuracy, these models are often opaque and difficult to interpret, which undermines trust and limits adoption in high-stakes applications. To address this gap, explainability for time series models has emerged as a critical area of research (Delaney et al., 2020; Bento et al., 2020). Existing methods primarily rely on saliency maps, which highlight time steps most associated with a predicted class (Håvardstun et al., 2024; S. et al., 2019; Doddaiah et al., 2022). Although useful, saliency-based explanations often lack human or higher-level interpretability, highlighting an input region is not equivalent to providing a meaningful rationale (Stubbin et al., 2024).

In computer vision, this limitation motivated concept-based explainability (Zhou et al., 2019; Bau et al., 2017; Boehmke & Greenwell, 2019), which aligns model predictions with human-understandable concepts such as 'stripes' or 'color'. By linking domain-relevant concepts to model gradients, these approaches have provided interpretable and thus more trustworthy insights in the context of vision tasks. Indeed, CAVs have been successfully deployed in real-world image applications (Kim et al., 2017), supported by pre-annotated concept banks (Zhou et al., 2019; Bau et al., 2017).

However, the extension of concept-based explainability to time series remains largely unexplored (Brenner et al., 2024; Madsen et al., 2023). A central obstacle is the absence of pre-annotated concept banks for temporal data: while vision benefits from large standardized sets of annotated concepts, time series domains lack comparable resources. Without structured concept banks, algorithms that can integrate high-level temporal concepts into deep learning classifiers remain underdeveloped.

Our work fills this critical gap by introducing a novel methodology for building time series concept banks and demonstrating their utility by incorporating the concepts into foundation models via adapter-based fine tuning. Our approach provides a principled foundation for concept-driven explainability

in time series modeling and ensures that models internalize higher-level (semantic, interpretable constructs derived from) temporal dynamics.

**Motivating Example.** Consider a deep time series classifier trained on ECG data to categorize patients into four classes: Arrhythmia, Sodium Blockade, Hemorrhage, and Cardiomyopathy. In high-stakes clinical settings, it is not sufficient for the model to output only class probabilities; physicians must also understand why the model favors one diagnosis over another.

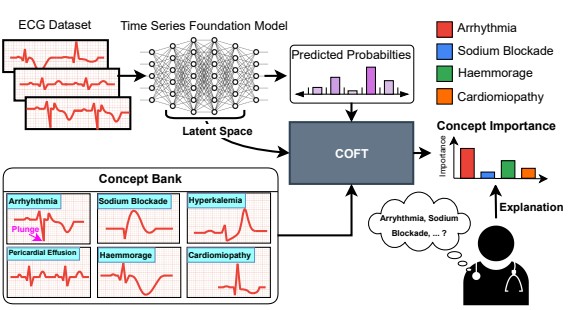

Figure 1: Our time series explainer method quantifies a model's sensitivity to specific ECG concepts. By creating a time series concept bank from diverse ECG datasets that captures essential ECG patterns, our approach evaluates whether the model utilizes these concepts in its predictions.

Current state-of-the-art approaches (Madsen et al., 2023; Ansari et al., 2024) provide predictions but often fail to reveal whether the classifier recognizes physiologically significant ECG patterns such as outbursts, plunges, or periodicities, shapes that domain experts associate with specific conditions. Our method addresses this limitation by constructing a concept bank and quantifying the model's sensitivity to these clinically relevant temporal concepts. For example, the presence of a plunge subsequence (shown in pink color) can increase the model's attention toward the Arrhythmia class, providing a transparent rationale and with that greater trust in the prediction. We illustrate this motivating example with Figure 1. An ECG snippet with key shapes (e.g., a plunge) highlighted in pink color. A corresponding concept attribution heatmap aligning the subsequence to the Arrhythmia class.

**Challenges.** Our problem is challenging for several reasons:

- *Defining Temporal Concepts*: In vision, concepts such as 'edges' or 'stripes' are visually intuitive and broadly understood. In contrast, time series concepts are often abstract like seasonality, drift, or domain-specific subsequences, making them harder to define, quantify, and ground in human intuition.

- *Dataset-Specificity*: Unlike vision, where concept banks (e.g., shapes, colors) are transferable across datasets, time series concepts are highly dataset-dependent. The absence of a standardized concept bank hinders consistent discovery, reuse, and benchmarking of temporal concepts across diverse applications.

- *Temporal Contextualization*: Time series are inherently sequential, with concept relevance shifting depending on temporal context and correlation across time steps. A subsequence that represents a concept in one region may lose significance in another, making robust concept extraction and alignment a nontrivial challenge.

- *Integrating with Low-Rank Adaptation*: While LoRA enables efficient fine-tuning of foundation models, it does not inherently enhance interpretability. A key challenge lies in devising mechanisms to inject high-quality temporal concepts into LoRA, ensuring that models internalize higher-level structures rather than merely optimizing predictive accuracy.

**Proposed Solution.** To address the above challenges, we propose COFT (Concepts for Foundation Time series models), a novel framework for concept-based explainability and fine-tuning in time series foundation models. COFT introduces a unified pipeline that (i) learns dataset-specific temporal concepts, (ii) derives concept-based explanations, and (iii) integrates these concepts directly into model training. At the core of COFT is the notion of time series concepts, derived from Shapelets, which capture maximally class-representative temporal subsequences. These subsequences are organized into high-quality dataset-tailored concept banks, enhancing the interpretability of model classifications by linking predictions to higher-level temporal patterns.

To further improve predictive performance, COFT incorporates these learned concepts into foundation models via adapter-based fine-tuning. This ensures that models not only recognize but also internalize

interpretable structures. COFT is model-agnostic and can be applied to a wide range of pre-trained architectures, including Fully Connected Networks, Transformers (Vaswani et al., 2017), and recent foundation models such as Amazon Chronos (Ansari et al., 2024) and MOMENT (Goswami et al., 2024). We validate COFT on the UCR time series benchmark, performing extensive hyperparameter and runtime analyses. Our results demonstrate that COFT yields higher-level concept-based explanations while also achieving significant accuracy improvements over state-of-the-art foundation time series models. This thus establishes a new pathway for interpretable and performant temporal modeling. Our main contributions are as follows:

- We introduce COFT, the first framework for systematically extracting temporal concepts, constructing a concept bank tailored to time series data.

- We extend the CAV methodology to sequential domains by developing metrics that evaluate whether models encode temporal concepts, detect their presence in specific subsequences, and quantify their causal influence on predictions.

- We show that integrating temporal concepts into adapter-based fine-tuning of foundation time series models not only enhances interpretability but also yields consistent gains in predictive accuracy across diverse benchmarks.

COFT is a unified framework; it extracts temporal concepts, uses LoRA to reinforce and operationalize the discovered concepts and generates CAV-based and TCAV-style explanations to verify whether models encode these concepts—making the representations causal, actionable, and explainable.

## 2 RELATED WORKS

Existing XAI approaches for time series can be categorized based on how they generate importance scores for model explanations:

**Saliency Maps.** These methods (Crabbe & van der Schaar, 2021; Parvatharaju et al., 2021) perturb input time series and measure changes in model outputs relative to a baseline to rank the importance of individual time steps. While useful, such maps are often non-intuitive, resulting in sub-optimal importance scores.

**Surrogate Models.** Methods like LIME (Ribeiro et al., 2016) and TSMULE (Schlegel et al., 2021; S. et al., 2019) train linear approximations locally to emulate a model's behavior for perturbed instances. When the relationship between perturbations and predictions is highly non-linear, as is common in deep time series models, these linear surrogates fail to provide meaningful explanations.

**SHAP Methods**. Shapley-based approaches (Bento et al., 2020; L. & L., 2017; Mujkanovic et al., 2020; Guillemé et al., 2019) estimate feature contributions by averaging the effects of random permutations. Extensions such as TIMESHAP adapt this to temporal data but still lack high-level, semantically meaningful explanations tied to human-understandable concepts.

**Concept-Based Explanations.** CAVs (Kim et al., 2017) and TCAV link abstract, human-understandable concepts (Zhou et al., 2019; Bau et al., 2017) to model outputs via gradients, providing interpretable reasoning in image classification (Ghorbani et al., 2019; Fong & Vedaldi, 2018). However, the sequential and domain-specific nature of time series makes the direct application of TCAV largely unexplored (Baur et al., 2020; Crabbe & van der Schaar, 2022; Brenner et al., 2024; Madsen et al., 2023; Yao et al., 2023).

Adapting TCAV to time series presents unique challenges: concepts must represent temporal subsequences, account for sequential dependencies, and align with domain-relevant patterns. Critically, existing methods do not improve pre-trained models or incorporate concept knowledge into fine-tuning. The integration of learned time series concept banks into foundation models thus remains an open and unexplored problem, one that motivates the development of methods like COFT.

## 3 METHODOLOGY

### 3.1 PROBLEM DEFINITION

We focus on the problem of generating concept-driven explanations for foundation time series models. Formally, assume we are given a dataset of $N$ time series $\mathcal{D} = X^1, \ldots, X^N$, where each sequence $X^i = [x_1^i, \ldots, x_T^i]$ lies in $\mathbb{R}^T$. Let $f_c : \mathbb{X} \to \mathcal{Y}$ denote a black-box classifier, where $\mathbb{X}$ is the $T$-dimensional feature space and $\mathcal{Y}$ the label space. For an instance of interest $X \in \mathcal{D}$ and a target class $C \in \mathcal{Y}$, our goal is to explain why $f_c(X)$ supports or rejects the prediction of class $C$.

Unlike attribution-based methods that highlight individual time points, we argue that interpretable explanations must operate at the level of subsequences that are semantically meaningful, recurring, and generalizable across samples. Our task is therefore to: (1) Identify class-discriminative subsequence-level concepts (2) quantify the model's sensitivity to these concepts, and (3) incorporate them into fine-tuning to improve both interpretability and performance.

### 3.2 THE PROPOSED METHOD: COFT

COFT is the first framework to systematically combine concept discovery, interpretability, and performance enhancement for foundation time series models. Given a trained model $f_c$ and a dataset $\mathcal{D}$, COFT operates in three tightly integrated stages: (1) The *Concept Bank*, which identifies high-quality Shapelets or critical patterns from time series data to develop domain-specific time series concepts. (2) These learned concepts are transformed into the model's activation space as *CAVs* and assessed for their presence in individual instances. (3) *Concept-Based Learning:* The model is enhanced by incorporating these learned concepts into training through adapter-based fine-tuning, leading to improved accuracy.

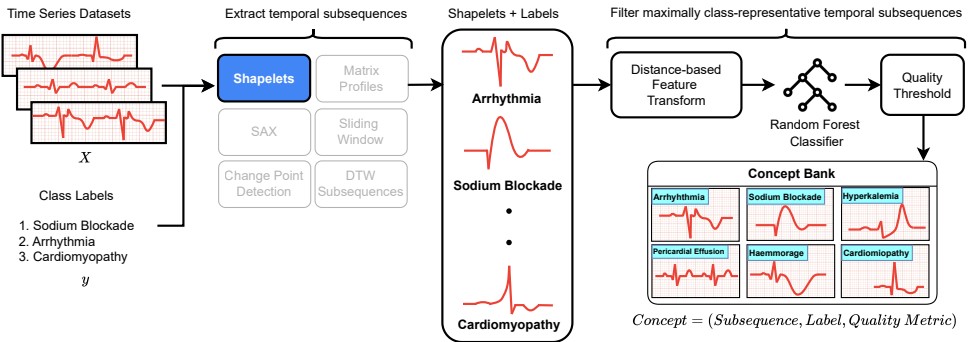

Figure 2: COFT Concept Bank Architecture

### 3.3 TIME SERIES CONCEPT BANK.

The first step of COFT is the extraction of candidate subsequences from raw time series data. We primarily use the Random Shapelet Transform (RST) (Guillaume et al., 2021), which samples subsequences of varying lengths and evaluates their class separation under dataset-specific hyperparameters, allowing COFT to capture patterns ranging from fine-grained motifs to broader structures. Although RST serves as the default extractor, COFT is modular: alternative methods such as Matrix Profiles, SAX, Change Point Detection, or sliding-window approaches can be substituted depending on dataset characteristics. With the shapelets extracted, we then apply convolution–based fast sequence matching to efficiently locate occurrences of the subsequence across the dataset. Given a time series $\mathbf{x} = [x_1, \ldots, x_T]$ and a candidate shapelet $\mathbf{s} = [s_1, \ldots, s_\ell]$, the convolution based peak detection is defined by:

$$\phi(\mathbf{x}, \mathbf{s}) = \arg\max \sum_{j=1}^{\ell} x_{t+j-1} \, s_{\ell-j+1}, \quad \text{for } t = 1, \ldots, T - \ell + 1 \tag{1}$$

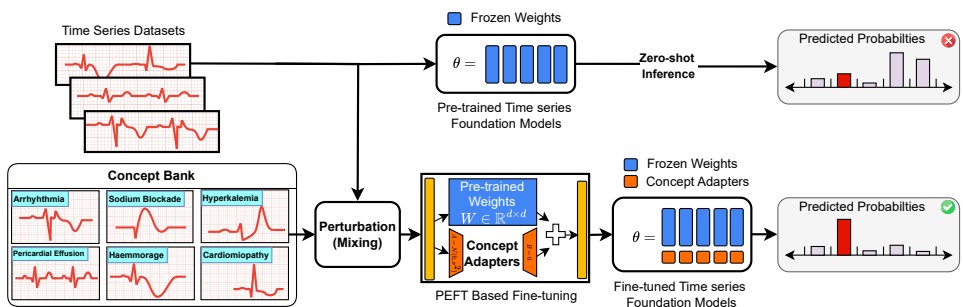

Figure 3: COFT Concept Adapter Architecture

A Random Forest classifier trained on these features provides a quantitative measure of concept quality:

$$Q(c) = \text{Acc}\Big( f_{\text{RF}}\big( f_{dt}(\phi(\mathbf{x}, \mathbf{s})) \big), y \Big) \tag{2}$$

where $\mathbf{x}$ is the time series, $\mathbf{s}$ is a candidate concept, $\phi(\mathbf{x}, \mathbf{s})$ denotes the peak locations of $\mathbf{x}$ with respect to $\mathbf{s}$, $f_{dt}$ is the distance based transform function (Algorithm 2) that computes distances between the closest matching subsequence of $x$ and shapelet $s$, $f_{\text{RF}}$ is the random forest classifier, $y$ are the ground-truth class labels, and $\text{Acc}(\cdot)$ is the classification accuracy used as the concept quality score. $f_{\text{RF}}$ handles noisy, redundant coordinates without tuning and provides an efficient, architecture-agnostic accuracy score for filtering high-quality concepts at scale. $f_{\text{RF}}$ is used only during concept filtering, not in the COFT model itself. This scoring procedure filters out noisy or spurious subsequences and retains only those strongly aligned with the target class. The resulting set of high-quality shapelets forms the concept bank—a repository of reusable, interpretable temporal patterns. Table 1 in the appendix compares multiple extractors and shows that RST yields the most discriminative and stable concepts, while the overall framework remains flexible to alternatives.

### 3.4 CONCEPT-BASED ENHANCED MODEL LEARNING.

Beyond interpretability, COFT leverages its concept bank to improve predictive performance of foundation models. This is achieved through concept-guided fine-tuning with Low-Rank Adapters(Hu et al., 2021) as shown in Figure 3. By incorporating the identified time series concepts into the pre-training of deep learning models, the research aims to enhance model performance and interpretability. To leverage the derived concepts in a way that improves model learning, COFT employs Concept-Based Fine-Tuning with Low-Rank Adaptation (Hu et al., 2021). This strategy is designed to be both parameter-efficient (avoiding full retraining of large models) and concept-aware (explicitly incorporating discovered subsequences into the fine-tuning process). The core idea is to generate a concept-perturbed training set, where each time series is augmented with subsequences from the concept bank, and then use these enriched samples to fine-tune the foundation model via LoRA adapters. The perturbation process begins with identifying where in a given time series a particular concept is expressed. For this purpose, COFT introduces a Fast Shapelet Matching procedure that combines $\phi(\mathbf{x}, \mathbf{s})$ with elastic sequence alignment as time series often exhibit warping or stretching. We refine the candidate regions using FastDTW, a lightweight approximation of Dynamic Time Warping (DTW), which measures elastic similarity:

$$\text{DTW}(\mathbf{x}, \mathbf{s}) = \min_{W} \left[ \sum_{k=1}^{L} euc(\mathbf{x}_{i_k}, \mathbf{s}_{j_k}) \right], d(\mathbf{x}_{t:t+\ell}, \mathbf{s}) = \text{DTW}(\mathbf{x}_{t:t+\ell}, \mathbf{s}) \tag{3}$$

where $euc(\mathbf{x}_{i_k}, \mathbf{s}_{j_k})$ is the eucliden pointwise distance and $W = \{(i_k, j_k)\}_{k=1}^{L}$ is a sequence of index pairs aligning elements of $\mathbf{x}$ to the element of $\mathbf{s}$. $W$ follows standard constraints: monotonicity and continuity (unit steps), ensuring indices increase without jumps. The subsequence window $[t, t + \ell)$ with the minimum distance is selected as the best match to the concept, provided that

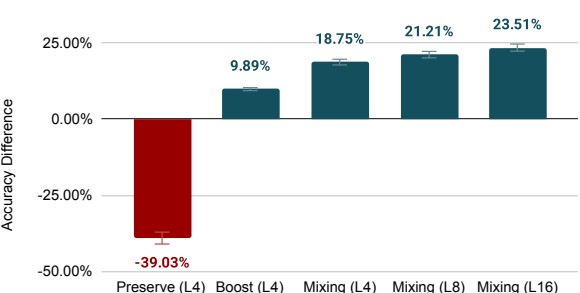

Figure 4: Study on Perturbation Strategies

$$d(\mathbf{x}_{t^*:t^*+\ell}, \mathbf{s}) \leq \tau \qquad (4)$$

where $\tau$ is a quality threshold ensuring only meaningful matches are perturbed. This two stage process balances efficiency with robustness to time warping. Once a suitable subsequence is identified, COFT applies one of three concept-guided perturbation strategies to generate synthetic training examples as shown in Figure 4 with LoRA ranks 4, 8 and 16:

- **Preserve strategy:** The time series is reduced to its salient subsequence by replacing the matched region with the concept shapelet and zeroing out the rest, producing clear, concept-isolated examples that highlight the shapelet's role in classification.

$$\mathbf{x}'_t = \begin{cases} \mathbf{s}_t, & t \in [t^*, t^* + \ell) \\ 0, & \text{otherwise} \end{cases} \qquad (5)$$

- **Boost strategy:** Scale the shapelet by a constant factor $\gamma > 1$ and replace only the matched subsequence. This strategy emphasizes the strength of the concept.

$$\mathbf{x}'_t = \begin{cases} \gamma \cdot \mathbf{s}_t, & t \in [t^*, t^* + \ell) \\ \mathbf{x}_t, & \text{otherwise} \end{cases} \qquad (6)$$

- **Mixing strategy:** The matched subsequence is blended with the original data using a mixing coefficient $\lambda \in [0, 1]$. This yields nuanced examples where the concept is present but not dominant, reflecting real-world cases with noise or overlapping patterns.

$$\mathbf{x}'_t = \begin{cases} \lambda \cdot \mathbf{s}_t + (1 - \lambda) \cdot \mathbf{x}_t, & t \in [t^*, t^* + \ell) \\ \mathbf{x}_t, & \text{otherwise} \end{cases} \qquad (7)$$

Collectively, the Preserve, Boost, and Mixing strategies produce a spectrum of concept-guided samples—from isolated to exaggerated to blended. Training on this enriched dataset with LoRA fine-tuning, COFT encourages the model to recognize concepts more reliably and generalize across their varied manifestations. Because LoRA updates only low-rank adapter layers while keeping the foundation model frozen, the approach remains both computationally and parameter efficient. COFT is described in Algorithm 1 in the Appendix.

### 3.5 TIME SERIES CONCEPT ACTIVATION VECTORS.

Once concepts are extracted, we measure their influence on model decisions using time series–specific CAVs. Unlike image-based CAVs, our adaptation accounts for temporal dependencies and sequential representations. Let $h(x)$ denote the hidden representation of input $x$ in model $f_c$. Using positive examples that contain concept $P$ and negative examples without it, we train a linear classifier in this representation space; its normal vector $v_P$ serves as CAV, capturing the latent direction associated with concept $P$:

$$w^* = \arg\min_w \frac{1}{n} \sum_{i=1}^{n} \ell(y_i, w^\top h(x_i)), \qquad v_P = \frac{w^*}{|w^*|} \qquad (8)$$

where $h(x_i) \in \mathbb{R}^d$ is the hidden representation of example $x_i$, $w$ is the trained weight vector of a linear classifier i.e concept direction learned from data, $\ell(\cdot, \cdot)$ is a classification (logistic) loss, $n$ is the number of examples used to train the linear classifier, and $y_i \in 0, 1$ indicates presence or absence of the concept. The sensitivity of $f_c$ to concept $P$ can then be quantified by measuring directional derivatives along $v_P$ (normalized Concept Activation Vector). High sensitivity indicates that the model's prediction is strongly influenced by the concept. This mechanism allows COFT to answer questions such as: Does the model rely on QRS peaks when predicting arrhythmia? Does it ignore small seasonal cycles in sales forecasting? Such insights go beyond attribution maps and provide concept-level interpretability.

## 4 EXPERIMENTS

### 4.1 EXPERIMENTAL SETUP

**Datasets** We evaluate the proposed COFT framework on benchmark datasets from the UCR Time Series Archive (Chen et al., 2015), a widely adopted repository for univariate time series classification, Non-UCR datasets are MIT-BIH Arrhythmia Database (Moody & Mark, 2001) and PTB ECG Datasets (Fu, 2021). All experiments follow the official train–test splits to ensure reproducibility

**Compared Methods** We introduce the first approach for constructing a time series concept bank and show how it can be integrated into both deep learning models and foundation models via LoRA. Prior time series XAI methods like TSMULE (Schlegel et al., 2021), TIMESHAP (Bento et al., 2020), LIME (Ribeiro et al., 2016), and PERT (Parvatharaju et al., 2021) focus on feature or instance-level explanations and do not provide subsequence- or concept-based interpretability, making direct comparison unsuitable. We evaluate COFT across multiple baselines and their concept-augmented counterparts. For Chronos (Ansari et al., 2024), we consider zero-shot (Chronos-Zero), LoRA-fine-tuned (Chronos-FT), and concept-integrated (Chronos-COFT) variants. Analogous baselines are defined for FCNs and Transformers. We further include CNNs, PatchTST (Nie et al., 2023), and MOMENT (Goswami et al., 2024) to demonstrate the extensibility and robustness of the COFT framework across diverse architectures.

**Implementation Details** Each baseline is trained on the designated training split and evaluated on the full test split. Architectures and hyperparameters follow their original implementations. For fine-tuning, we employ LoRA using the PEFT library (Xu et al., 2023). Experiments are implemented in PyTorch using the Adam optimizer with standard learning-rate decay. For PEFT, we use a model- and dataset-agnostic LoRA rank of 4, resulting in 1–3% additional trainable parameters. Code is made available https://anonymous.4open.science/r/adapt_concept_tsft-03DD

**Metrics** *1) Concept-based Interpretability.* We use the TCAV metric (Ghorbani et al., 2019) to quantify model sensitivity to learned time-series concepts, removing reliance on subjective human assessment and providing a model-centric view of decision boundaries. We assess causal importance using SSC and SDC, which measure the minimal concept sets needed to preserve or disrupt correct predictions. *2) Model Performance.* We report *accuracy, precision, recall* and *F1* before and after fine-tuning. COFT improves all metrics, indicating that concept-guided adaptation enhances interpretability while steering models toward more informative subsequences. Performance gains are most pronounced with high-quality concepts, affirming the role of well-defined temporal patterns in effective model learning.

### 4.2 COFT DISCOVERS CLASS-SPECIFIC CONCEPTS.

To illustrate the class-specific temporal patterns COFT identifies, we show representative subsequences from the MIT-BIH Arrhythmia dataset in Figure 5. Each panel displays a characteristic ECG pattern along with its concept quality score (Q). The waveforms illustrate clear class-specific differences—ranging from irregular oscillations to sharp spikes and broad excursions—highlighting the discriminative temporal structures captured by COFT. Additional examples are available in Appendix A.3.2.

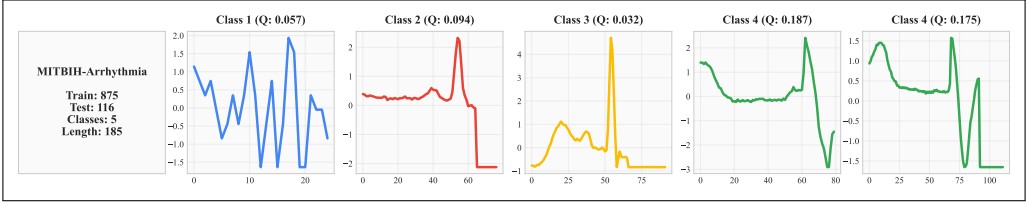

Figure 5: MIT-BIH Arrhythmia concepts; colors denote class labels and Q is a quality measure

To address the challenge of generating explanations that are faithful to the model's internal decision process and comprehensible to humans, we rank concepts using their TCAV scores.

Intuitively, this ranking serves two purposes. 1) It reflects the degree to which each concept influences the classifier's predictions, thereby ensuring local faithfulness to the model's behavior. 2) Presenting concepts in a ranked order makes the explanation more cognitively aligned with human reasoning. Humans naturally prioritize information by salience, focusing first on the most influential or recognizable factors before considering less relevant ones.

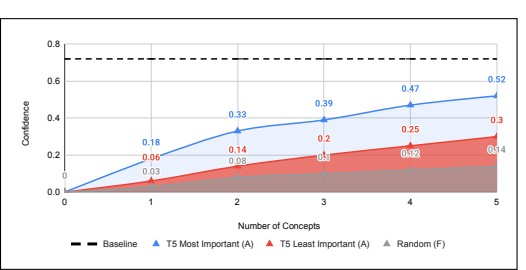

Figure 6: Study on Smallest Sufficient Concept

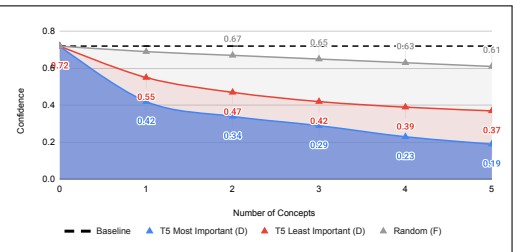

Figure 7: Study on Smallest Deleted Concept

To validate these rankings, we use Smallest sufficient concepts (SSC) and Smallest destroying concepts (SDC) (Ghorbani et al., 2019) analyses, which compare concept-level perturbations against feature-level perturbations. In our setup, TCAV-ranked perturbations correspond to concept-level insertions or deletions, while the Random condition corresponds to feature-level manipulations at the timestep level. For *insertion*, we start from an opposing-class instance and iteratively replace subsequences (or individual timesteps) with those from the target class, tracking the increase in confidence. For *deletion*, we begin with a target-class instance and iteratively replace its subsequences or timesteps with those from the opposing class, measuring the decline in confidence. This preserves data distribution and avoids artifacts introduced by naïve zeroing. As shown in Figures 6 and 7, feature-level perturbations (Random) induce only small, inconsistent changes in confidence, reflecting their brittleness.

In contrast, perturbing higher-ranked TCAV concepts produces large, monotonic, and class-consistent effects—confidence rises sharply in SSC and drops sharply in SDC. These results demonstrate that COFT identifies concepts with genuine semantic influence and that concept-level perturbations provide substantially more informative and stable explanations than feature-level ablations.

### 4.3 COFT IMPROVES MODEL ACCURACY.

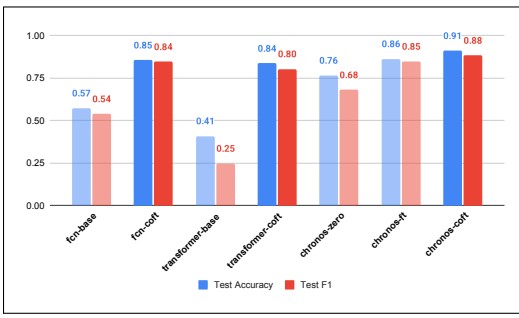

Figure 8: Baselines vs. COFT F1 and accuracy across FCN, Transformer, and Chronos models on UCR datasets

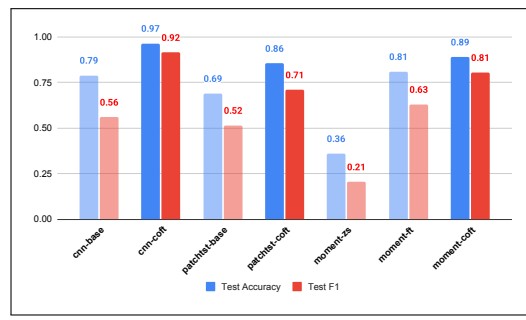

Figure 9: Baselines vs. COFT F1 and accuracy across CNN, PatchTST, and Moment FM on the MIT-BIH Arrhythmia and PTB-DB datasets

Fine-tuning with learned concepts systematically improves accuracy by steering models toward informative subsequences and away from spurious patterns. Across many UCR datasets, our method consistently outperforms conventional approaches, as reported in the Appendix Table 2. Across all architectures, COFT consistently improves performance by integrating semantically coherent time-series concepts during fine-tuning. Chronos-COFT delivers clear and statistically significant gains over both Chronos-Zero and Chronos-FT, and the same pattern holds for FCN-COFT and

Transformer-COFT, which reliably outperform their zero-shot and naïve LoRA counterparts. ==These trends extend across diverse settings: CNN-COFT achieves 0.96 accuracy on PTB-ECG (vs. 0.75 for CNN-Base), FCN-COFT reaches 1.00 accuracy on UCR datasets such as Proximal Phalanx TW, and PatchTST-COFT raises accuracy from 0.30 to 0.97 on ArrowHead.== Foundation models such as Moment and Chronos show similar gains, demonstrating that concept-guided adaptation improves precision, recall, and F1 across both UCR benchmarks and non-UCR clinical datasets. These gains underscore COFT's robustness and extensibility across models and domains.

### 4.3.1 CASE STUDY: EEG SLEEP SLEEP DATASET

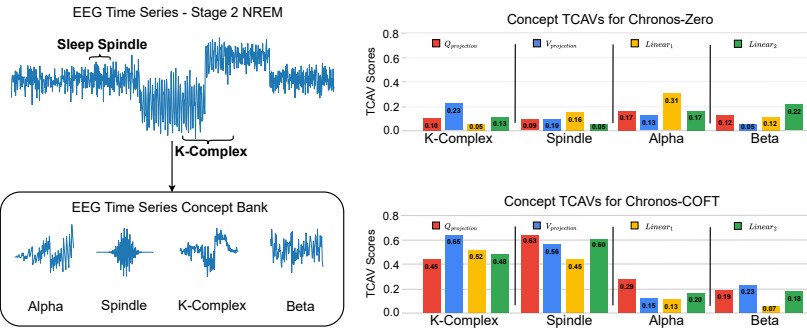

Figure 10: EEG Sleep (Kemp et al., 2000) Case study.

We present a case study conducted in collaboration with clinical sleep experts from the XX Medical School[1] to evaluate the interpretability of our approach on the EEG Sleep dataset (Kemp et al., 2000). As illustrated in Figure 10, COFT constructs an EEG-specific concept bank that captures clinically salient temporal patterns, including *Alpha*, *Beta*, *K-complex*, and *Sleep Spindle* activity. Incorporating these learned concepts during fine-tuning not only increases the foundation model's confidence in recognizing EEG patterns but also improves its predictive reliability. Notably, the fine-tuned Chronos-COFT model assigns substantially higher importance to K-complex and Spindle concepts relative to the zero-shot Chronos baseline model as reflected by the TCAV sensitivity scores across layers such as $Q_{projection}$, $V_{projection}$, $Linear_1$, and $Linear_2$. This enhanced alignment between model attributions and clinically validated EEG markers demonstrates that COFT enables foundation models to internalize human-interpretable concepts, thereby improving both performance and transparency beyond what is achievable with traditional foundation models.

## 5 CONCLUSION

In this work, we identify time series concepts as a critical foundation for generating human-interpretable explanations of time series classifiers. We propose COFT, a novel framework that discovers and leverages discriminative subsequences to improve both interpretability and predictive performance. A central contribution is the new adaptation of CAVs (Ghorbani et al., 2019) to the time series domain, enabling the construction of a concept bank that captures essential temporal patterns. These learned concepts are then incorporated into foundation models through parameter-efficient fine-tuning with LoRA, enhancing model accuracy without compromising scalability. We validate COFT on the UCR and Non-UCR univariate time series benchmark, evaluating across diverse architectures including fully connected networks, Transformers, CNN, PatchTST, and foundation models such as Amazon Chronos (Ansari et al., 2024) and Moment (Goswami et al., 2024). Experimental results show that COFT provides: (1) Higher-level explanations, quantified by CAV-based metrics, and (2) Performance improvements, where LoRA fine-tuning guided by learned concepts consistently boosts accuracy. Together, these findings position COFT as a robust framework that unifies interpretability and predictive performance in time-series modeling.

---

[1] Name of Medical School omitted for anonymity.

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

# A  APPENDIX

## A.1  ALGORITHM.

---

**Algorithm 1:** Shapelet-Guided LoRA Fine-Tuning Pipeline (COFT)

---

**Input:** UCR dataset $\mathcal{D} = (X_{\text{train}}, y_{\text{train}}, X_{\text{test}}, y_{\text{test}})$
**Output:** Trained LoRA model $M_{\text{LoRA}}$
**Stage 1: Base Model Training**
1: Train a deep neural network $M_{\text{base}}$ on $(X_{\text{train}}, y_{\text{train}})$
**Stage 2: Shapelet Extraction**
2: Fit a Random Shapelet Transform (RST) on $(X_{\text{train}}, y_{\text{train}})$
3: Extract shapelets $S = \{s_1, \dots, s_k\}$ from RST
4: Obtain transformed representations $X_{\text{train}}^{\text{shp}}, X_{\text{test}}^{\text{shp}}$
**Stage 3: Shapelet Quality Scoring**
5: Train a Random Forest on $X_{\text{train}}^{\text{shp}}$ and compute quality scores $Q(s)$ for each shapelet $s \in S$
**Stage 4: Shapelet Matching and Augmentation**
6: For each instance $x$, find closest shapelet via fast matching $\phi_{\text{match}}(x, s)$
7: Generate synthetic examples $(X_{\text{cand}}, y_{\text{cand}})$ using perturbation guided by $S$
**Stage 5: LoRA Fine-Tuning**
8: Construct $M_{\text{LoRA}}$ by freezing $M_{\text{base}}$ and inserting low-rank adapters
9: Fine-tune $M_{\text{LoRA}}$ on augmented data $(X_{\text{train}} \cup X_{\text{cand}},\ y_{\text{train}} \cup y_{\text{cand}})$
**return** $M_{LoRA}$

---

---

**Algorithm 2:** Fast Shapelet Matching

---

**Input:** Time series $t$, shapelet $s$
**Output:** Minimum DTW distance $d$, best-match interval $(i^*, j^*)$
$d \leftarrow \infty, \quad index \leftarrow null$
Compute convolution: $c \leftarrow ts * \text{flip}(s)$
Identify candidate peaks: $P \leftarrow \{i \mid c_i = \max(c)\}$
**foreach** $i \in P$ **do**
  **if** $|t[i : i + |s|]| = |s|$ **then**
    $\_d \leftarrow \text{DTW}(t[i : i + |s|],\ s)$
    **if** $\_d < d$ **then**
      $d \leftarrow \_d$
      $index \leftarrow (i,\ i + |s|)$

**return** $(d,\ index)$

---

## A.2  ABLATION STUDY.

### A.2.1  IMPACT OF MODEL SIZE.

The Amazon Chronos Ansari et al. (2024) and MOMENT Goswami et al. (2024) foundational language models are available in various sizes, including Tiny, Small, Mini, Base, and Large. To evaluate their performance, we ran on all 128 UCR time series classification datasets across these different model sizes. Our analysis revealed that the accuracy achieved by the models was fairly consistent across all sizes, with only slight variations as shown in Figure 11. That is, the Large size models demonstrated a marginal improvement in accuracy compared to the other sizes. Based on these findings, we had decided to use the Large model size for experiments in this paper to ensure optimal performance while maintaining consistency. Further, we observed that Amazon Chronos Ansari et al. (2024) outperformed MOMENT Goswami et al. (2024) in overall accuracy across the 128 UCR datasets. Therefore, Amazon Chronos was chosen for overall experiments.

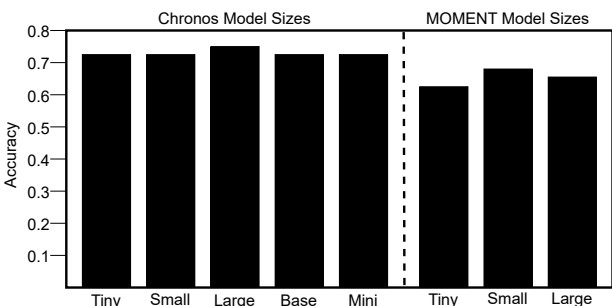

Figure 11: Hyperparameter Study: Accuracy of Different Model Sizes Across all 128 UCR Chen et al. (2015) Datasets.

### A.2.2 COMPARING SHAPELET EXTRACTION METHODS.

To identify valuable subsequences, which form the foundational elements of concept discovery in time series data, we conducted an extensive study using several state-of-the-art Shapelet extraction methods from open-source libraries Lines et al. (2012); Ye & Keogh (2009). These methods included Random Shapelet Transform, Fast Shapelet, and Learning Shapelet, each offering different strengths in terms of speed, accuracy, and interpretability. We evaluated each method using quantitative metrics, including classification accuracy, interpretability of the extracted Shapelets, and computational efficiency. Based on these metrics, we selected Random Shapelet Transform (RST) as the optimal technique for extracting Shapelets across all our datasets as shown in the Table 1.

This Random Shapelet Transform method outperformed others in terms of its ability to capture diverse subsequences while maintaining a balance between computational costs and the clarity of the resulting Shapelets. For each dataset, however, the performance of Random Shapelet Transform was highly dependent on proper hyperparameter tuning. To derive the most informative and interpretable Shapelets, we adjusted key parameters such as the number of Shapelets, their length, and the regularization factors. This tuning process was essential for ensuring that the extracted Shapelets captured the most relevant features of the data without overfitting or generating noise. In particular, we focused on identifying Shapelets that were not only predictive but also aligned with human-understandable concepts, with the later critical for the broader goal of concept-based model interpretability.

| METHOD | ELAPSED TIME (SECS) | ACCURACY ($Q$) | INTERPRETABILITY |
|---|---|---|---|
| Fast Shapelets | 6.91 ± 10.14 | 84.07% ± 0.16 | Medium (explicit shapelets) |
| Learning Shapelet | 66.70 ± 33.73 | 81.11% ± 0.10 | Low - Medium (latent space, harder to interpret) |
| RST | **4.92 ± 2.57** | **88.43% ± 0.09** | High (very explicit, human-readable) |

Table 1: Shapelet extraction method study. Here, RST stands for Random Shapelet Transform.

The RST automatically validates the quality of Shapelets by evaluating their impact on classification accuracy. Additionally, we generated synthetic datasets with predefined patterns and utilized widely recognized real-world datasets containing well-defined domain-specific Shapelets, such as Apache Spark Jacob et al. (2020) and Gunpoint Chen et al. (2015). These datasets enabled manual validation of the quality of the generated Shapelets across various Shapelet extraction methods.

### A.2.3 ROBUSTNESS UNDER VARYING CONCEPT QUALITY.

To evaluate the robustness of our approach, we systematically vary the quality of the concept inputs used during fine-tuning and measure the resulting changes in model performance. Specifically, we perturb the quality of extracted subsequences by controlling their alignment with ground-truth

concepts as shown in the Figure 12. Concept quality is quantified using a similarity score ranging from 0.7 to 0.95, where higher values indicate stronger alignment with expert-validated subsequences.

These values are reported on the x-axis, while the corresponding F1 scores are shown on the y-axis. Across all regimes, Chronos-COFT consistently outperforms the zero-shot Chronos baseline, underscoring its ability to remain effective even when concept inputs are degraded.

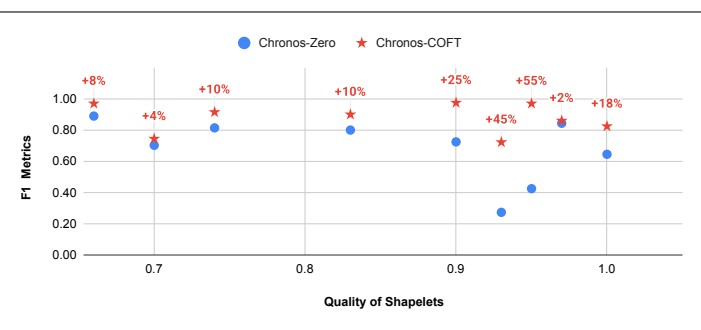

Figure 12: Study on robustness of subsequences

**High Quality (0.90–0.95)** The performance divergence is most pronounced in this regime: Chronos-COFT delivers large relative improvements, with gains increasing from +25% to +55% over the baseline.

**Moderate Quality (0.70–0.90)** Improvements are more modest but stable, typically ranging from +4% to +10%.

**Ceiling Effect** When the baseline F1 score is already near its upper bound, the relative advantage of COFT narrows to +2% to +4%.

This analysis highlights that our method is robust to perturbations in concept quality: even when concepts are noisy or imperfect, COFT yields consistent improvements, while higher-quality concepts amplify the relative gains.

## A.3 EXPERIMENTAL RESULTS

Refer to the next page for table and chart details.

Across both the non-UCR (PTB ECG, MIT-BIH Arrhythmia) and UCR benchmark datasets, the COFT-enhanced models consistently outperform their base and fine-tuned counterparts across all metrics accuracy, precision, recall, and F1 score, demonstrating the effectiveness of the COFT approach. Accuracy measures the overall proportion of correct predictions. Precision captures how often predicted positives are correct, while recall measures how many true positives are detected. The F1-score combines precision and recall into a single balanced metric, especially useful under class imbalance. For classical deep learning models such as CNNs or FCNs, COFT yields substantial improvements; for instance, CNN-COFT achieves 0.96 accuracy on PTB ECG compared to 0.75 for CNN-Base, while FCN-COFT reaches up to 1.00 accuracy on certain UCR datasets like Proximal Phalanx TW. Transformer-based models similarly benefit, with PatchTST-COFT and Transformer-COFT showing large gains over their respective base models, particularly in datasets where the base transformer struggles (e.g., ArrowHead: 0.30 to 0.97). Timeseries Foundation Models also see consistent metric boosts: Moment-COFT or Chronos-COFT often surpass both zero-shot and fine-tuned variants, suggesting that COFT provides a robust task-adaptive enhancement. Patterns across datasets indicate that COFT is especially effective for models or tasks that initially underperform low base accuracy, precision, or recall is greatly increased, while already strong models receive smaller but still meaningful improvements. Overall, COFT improves both model reliability (precision and recall) and balanced performance (F1), making it a broadly effective fine-tuning strategy across diverse timeseries datasets.

### A.3.1 COFT IMPROVES ACCURACY ACROSS NEW MODELS.

| DATASET | MODEL | METHOD | METRICS | | | |
|---|---|---|---|---|---|---|
| | | | ACCURACY | PERCISION | RECALL | F1 |
| Arrow Head | Deep Neural Network | fcn-base | 0.53 | 0.63 | 0.54 | 0.58 |
| | | fcn-coft | 0.94 | 0.94 | 0.94 | 0.94 |
| | Timeseries Transformer | transformer-base | 0.30 | 0.10 | 0.33 | 0.15 |
| | | transformer-coft | 0.97 | 0.97 | 0.97 | 0.97 |
| | Timeseries Foundation Model | chronos-zero | 0.73 | 0.66 | 0.75 | 0.70 |
| | | chronos-ft | 0.74 | 0.83 | 0.85 | 0.84 |
| | | chronos-coft | 0.80 | 0.87 | 0.65 | 0.74 |
| ECG200 | Deep Neural Network | fcn-base | 0.84 | 0.85 | 0.80 | 0.82 |
| | | fcn-coft | 1.00 | 1.00 | 1.00 | 1.00 |
| | Timeseries Transformer | transformer-base | 0.64 | 0.32 | 0.50 | 0.39 |
| | | transformer-coft | 0.97 | 0.98 | 0.96 | 0.97 |
| | Timeseries Foundation Model | chronos-zero | 0.92 | 0.91 | 0.87 | 0.89 |
| | | chronos-ft | 1.00 | 1.00 | 1.00 | 1.00 |
| | | chronos-coft | 1.00 | 1.00 | 1.00 | 1.00 |
| Fish | Deep Neural Network | fcn-base | 0.13 | 0.02 | 0.14 | 0.03 |
| | | fcn-coft | 0.82 | 0.73 | 0.82 | 0.77 |
| | Timeseries Transformer | transformer-base | 0.13 | 0.02 | 0.14 | 0.03 |
| | | transformer-coft | 0.23 | 0.07 | 0.24 | 0.11 |
| | Timeseries Foundation Model | chronos-zero | 0.84 | 0.89 | 0.75 | 0.81 |
| | | chronos-ft | 0.85 | 0.90 | 0.87 | 0.88 |
| | | chronos-coft | 0.89 | 0.91 | 0.92 | 0.91 |
| Gun Point | Deep Neural Network | fcn-base | 0.68 | 0.70 | 0.68 | 0.69 |
| | | fcn-coft | 0.97 | 0.97 | 0.97 | 0.97 |
| | Timeseries Transformer | transformer-base | 0.49 | 0.25 | 0.50 | 0.33 |
| | | transformer-coft | 0.67 | 0.80 | 0.66 | 0.73 |
| | Timeseries Foundation Model | chronos-zero | 0.96 | 0.88 | 0.81 | 0.84 |
| | | chronos-ft | 0.97 | 0.88 | 0.83 | 0.85 |
| | | chronos-coft | 0.97 | 0.86 | 0.86 | 0.86 |
| Italy PowerDemand | Deep Neural Network | fcn-base | 0.96 | 0.96 | 0.96 | 0.96 |
| | | fcn-coft | 1.00 | 1.00 | 1.00 | 1.00 |
| | Timeseries Transformer | transformer-base | 0.50 | 0.25 | 0.50 | 0.33 |
| | | transformer-coft | 0.99 | 0.99 | 0.99 | 0.99 |
| | Timeseries Foundation Model | chronos-zero | 0.97 | 0.75 | 0.70 | 0.72 |
| | | chronos-ft | 0.96 | 0.97 | 0.94 | 0.95 |
| | | chronos-coft | 0.96 | 0.99 | 0.96 | 0.97 |
| Proximal Phalanx TW | Deep Neural Network | fcn-base | 0.35 | 0.06 | 0.17 | 0.09 |
| | | fcn-coft | 1.00 | 1.00 | 1.00 | 1.00 |
| | Timeseries Transformer | transformer-base | 0.35 | 0.06 | 0.17 | 0.09 |
| | | transformer-coft | 0.89 | 0.77 | 0.78 | 0.77 |
| | Timeseries Foundation Model | chronos-zero | 0.70 | 0.79 | 0.81 | 0.80 |
| | | chronos-ft | 0.85 | 0.79 | 0.83 | 0.81 |
| | | chronos-coft | 0.91 | 0.89 | 0.91 | 0.90 |
| Shapelet Sim | Deep Neural Network | fcn-base | 0.54 | 0.55 | 0.54 | 0.55 |
| | | fcn-coft | 0.58 | 0.60 | 0.58 | 0.59 |
| | Timeseries Transformer | transformer-base | 0.50 | 0.25 | 0.50 | 0.33 |
| | | transformer-coft | 1.00 | 1.00 | 1.00 | 1.00 |
| | Timeseries Foundation Model | chronos-zero | 0.46 | 0.25 | 0.30 | 0.27 |
| | | chronos-ft | 0.58 | 0.55 | 0.52 | 0.53 |
| | | chronos-coft | 0.71 | 0.68 | 0.77 | 0.72 |
| Symbols | Deep Neural Network | fcn-base | 0.63 | 0.75 | 0.63 | 0.68 |
| | | fcn-coft | 0.79 | 0.73 | 0.78 | 0.76 |
| | Timeseries Transformer | transformer-base | 0.34 | 0.13 | 0.33 | 0.18 |
| | | transformer-coft | 1.00 | 1.00 | 1.00 | 1.00 |
| | Timeseries Foundation Model | chronos-zero | 0.56 | 0.41 | 0.44 | 0.42 |
| | | chronos-ft | 1.00 | 1.00 | 1.00 | 1.00 |
| | | chronos-coft | 1.00 | 1.00 | 1.00 | 1.00 |
| Trace | Deep Neural Network | fcn-base | 0.49 | 0.38 | 0.52 | 0.44 |
| | | fcn-coft | 0.58 | 0.53 | 0.60 | 0.57 |
| | Timeseries Transformer | transformer-base | 0.43 | 0.31 | 0.50 | 0.38 |
| | | transformer-coft | 0.80 | 0.65 | 0.74 | 0.69 |
| | Timeseries Foundation Model | chronos-zero | 0.74 | 0.66 | 0.63 | 0.64 |
| | | chronos-ft | 0.81 | 0.75 | 0.74 | 0.74 |
| | | chronos-coft | 0.95 | 0.80 | 0.85 | 0.82 |

Table 2: Results for FCN-Base, FCN-COFT, Transformer-Base, Transformer-COFT, Chronos-Zero, Chronos-FT, and Chronos-COFT across the UCR benchmark datasets.

| DATASET | MODEL | METHOD | METRICS | | | |
|---|---|---|---|---|---|---|
| | | | ACCURACY | PERCISION | RECALL | F1 |
| PTB ECG | Convolutional Neural Network | CNN-Base
CNN-COFT | 0.75
0.96 | 0.76
0.94 | 0.83
0.97 | 0.74
0.95 |
| | PatchTST | PatchTST-Base
PatchTST-COFT | 0.51
0.75 | 0.47
0.70 | 0.47
0.71 | 0.46
0.70 |
| | Timeseries Foundation Model | Moment-ZeroShot
Moment-FT
Moment-COFT | 0.71
0.72
0.82 | 0.35
0.71
0.80 | 0.49
0.77
0.88 | 0.41
0.70
0.81 |
| MIT-BIH Arrhythmia | Convolutional Neural Network | CNN-Base
CNN-COFT | 0.83
0.97 | 0.37
0.96 | 0.42
0.83 | 0.38
0.88 |
| | PatchTST | PatchTST-Base
PatchTST-COFT | 0.87
0.96 | 0.59
0.75 | 0.57
0.70 | 0.57
0.72 |
| | Timeseries Foundation Model | Moment-ZeroShot
Moment-FT
Moment-COFT | 0.01
0.90
0.96 | 0.01
0.56
0.82 | 0.2
0.60
0.79 | 0.0
0.56
0.80 |

Table 3: Results for CNN-Base, CNN-COFT, PatchTST-Base, PatchTST-COFT, Moment-ZeroShot(ZS), Moment-FineTuning(FT), and Moment-COFT across the non-UCR benchmark datasets (MIT-BIH Arrhythmia Database and PTB Diagnostic ECG Database).

### A.3.2 TEMPORAL STRUCTURE ACROSS DATASETS

Across the UCR datasets we use from short physiological signals (ECG200) to gesture-like traces (GunPoint), longer heterogeneous series (Fish), and synthetic shapelet-driven data (ShapeletSim) Figure 13 illustrates the range of localized patterns and variability encountered in practice. We focus on shapelets, short subsequences that capture the discriminative regions of a time series and align well with the kinds of features practitioners naturally look for, such as the peak in GunPoint or the abrupt excursions in MITBIH-Arrhythmia. Their locality and robustness to small temporal shifts make them well-suited to these datasets. While our experiments center on shapelets, the same approach can be extended to other interpretable primitives change points, SAX patterns, or simple spectral cues setting the stage for a broader concept bank in future work.

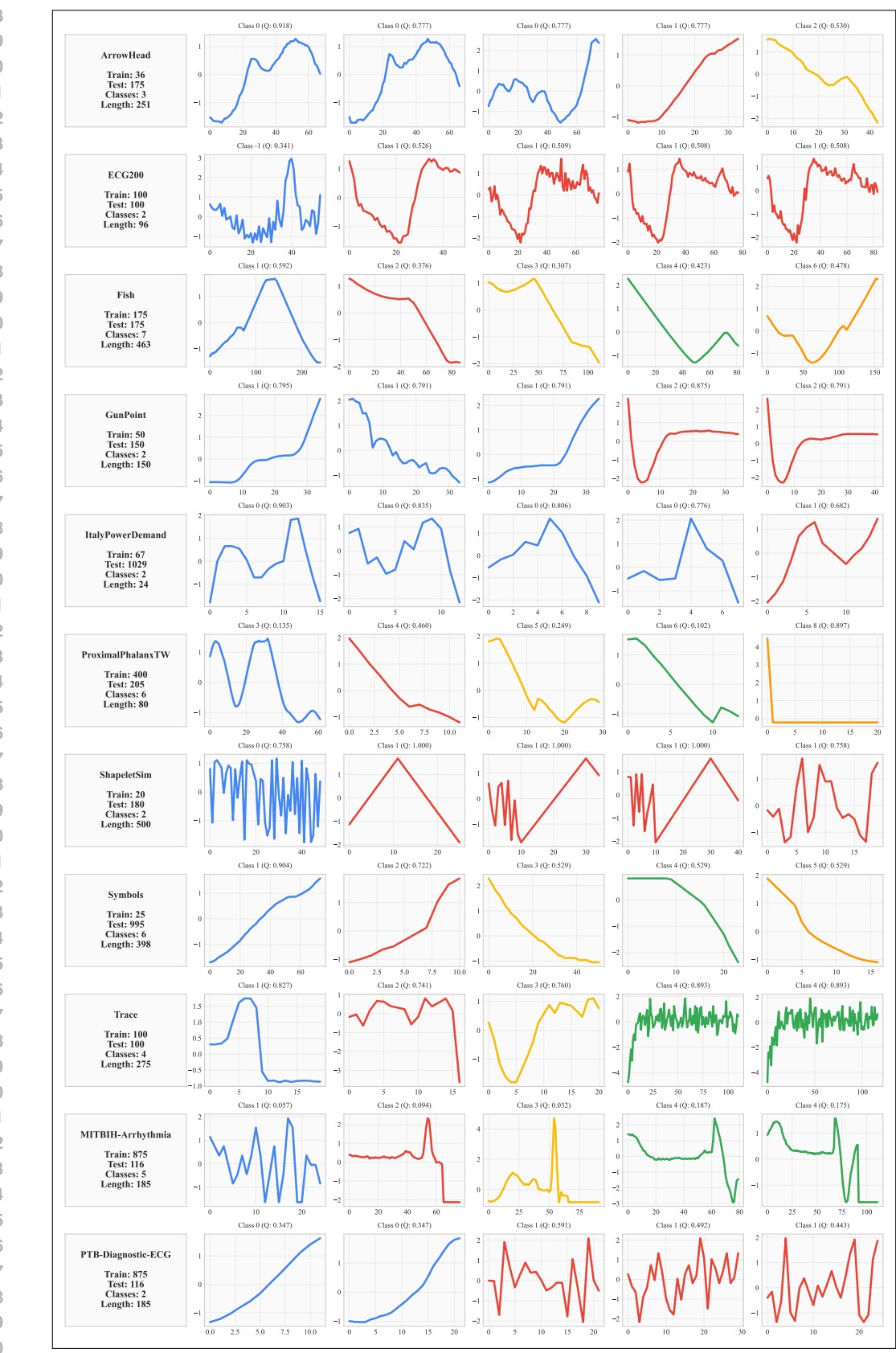

Figure 13: Dataset overview with corresponding concepts; colors denote class labels and $Q$ is a quality measure

