# OpenReview forum: "Adapting and Explaining Time Series Foundation Models with Concepts"
_ICLR.cc/2026/Conference — Submitted to ICLR 2026_

### Official Review · Reviewer_ryLn · 2025-10-20

**Soundness:** 2
**Presentation:** 2
**Contribution:** 2
**Rating:** 2
**Confidence:** 4

**Summary:**

The paper proposes COFT, a framework to adapt Concept Activation Vectors (CAVs) for time series foundation models by discovering and organizing temporal concepts through shapelet-based transformations. COFT applies Low-Rank Adaptation (LoRA) for efficient integration of concepts in models during fine-tuning.

**Strengths:**

Strengths:
- The proposed method is novel and potentially impactful to improve model accuracy and interpretability
- The paper includes both zero-shot and fine-tuned baselines for several models. The inclusion of tine-tuning baselines without COFT is valuable.
- The proposed method achieves better accuracy than the compared baselines.
- he authors use specific healthcare examples and patterns to motivate and contextualize their work.

**Weaknesses:**

Weaknesses:
- Limited baseline architectures. The baselines include only one time series foundation model (Chronos), despite the paper’s focus on embedding concepts in time series foundation models. This theme is explicitly highlighted in the title. Although Moment is frequently discussed in the text, it is not included in the empirical evaluation. Given that Moment is trained on the UCR/UEA classification archive (train set), its inclusion would be relevant.
- Lack of representative baselines. The chosen baselines, a vanilla fully connected network (FCN) and a vanilla Transformer, are not the most competitive models commonly used with time series. Including additional baselines such as CNN-based models or patch-based tokenization Transformers (e.g., PatchTST) would better situate this work within the context of actively used models in practice.
- No comparison with non–deep learning methods. Important classical baselines such as majority class prediction, random forecast, or XGBoost are not included. These comparisons would provide a clearer understanding of the relative gains from the proposed approach.
- Limited evaluation data. The evaluation appears to rely primarily on the general UCR database. Considering additional classification benchmarks would strengthen the empirical analysis.
- Lack of error measures. Figure 7 does not report variability (e.g., standard deviation or confidence intervals). Repeating experiments with multiple random seeds would help assess the robustness and variance of model performance.

Other Comments:
- Developing an open-source concept bank derived from real-world time series (excluding TSFM benchmark/test datasets) would be highly valuable and could provide a readily available resource for fine-tuning open-source TSFMs.
- Line 294: The figure caption formatting appears inconsistent with the main text. It should be properly spaced from the surrounding paragraphs.
- Extending the concept framework beyond medical data to domains such as finance could be compelling. For example, “flag” patterns are often used as indicators in financial time series.
- Including a summary table listing each dataset’s name, domain, and the specific patterns (or subset of examples) extracted would improve clarity. Broadening the scope to include additional domains such as finance and energy would also enhance the paper’s impact.

**Questions:**

Questions:
- Lines 256–257: “A random forest classifier is trained on these features, and its classification accuracy provides a quantitative quality measure. This score is used to filter and retain only the most representative concepts.” How exactly is this score used for filtering? Is there a specific threshold applied to determine which concepts are retained?
- Are the concepts extracted exclusively from the training sets of the datasets, or from external datasets? This distinction is important—concepts should not be extracted from the test sets used for evaluation.

---

> ### Author Response · Authors · 2025-11-26
> **Response to reviewer ryLn**
>
> Thank you for the opportunity to clarify; we are confident the points below comprehensively address your questions.
>
> > W1) Only one foundation model (Chronos). Moment discussed but not included.
>
> `R: We appreciate the concern and clarify that Moment did not support PEFT at the time of our original submission.` Since the addition of PEFT support, we have now included new experiments using the Moment foundation model on non-UCR datasets (MIT-BIH Arrhythmia, PTB-ECG). Section A.3.1 (Figure 11) in the Appendix now reports results comparing CNN-Base vs. CNN-COFT, PatchTST-Base vs. PatchTST-COFT, Moment-ZS, Moment-FT, and Moment-COFT, and Figure 11 demonstrates that COFT yields consistent gains across these new model families and domains. These results confirm that COFT is fully model-agnostic, integrates seamlessly with any Time Series Foundation model or deep learning architecture, and that concept-guided fine-tuning systematically improves accuracy by reinforcing informative subsequences, including on entirely new models and datasets as suggested.
>
> > W2) Baselines not representative; include CNNs or PatchTST.
>
> `R: Thank you for the suggestion. We now include CNN, PatchTST, and MomentFM evaluations on non-UCR datasets to demonstrate the ease of extending COFT.` Specifically, we compare CNN-Base vs. CNN-COFT, PatchTST-Base vs. PatchTST-COFT, and add MomentFM in its zero-shot, fine-tuned, and COFT variants. As shown in Section A.3.1 (Fig. 11), COFT consistently improves performance across convolutional, patch-based, and foundation-model encoders, reinforcing that the method is architecture-agnostic and robust across diverse time-series models.
>
> > W3) No comparison with classical (majority/random/XGBoost) baselines.
>
> `R: We agree that classical baselines such as majority, random, or XGBoost are valuable in general benchmarking, but are not meaningful comparators in our setting.` COFT works by intervening on concept directions in the model’s latent space and adjusting those directions through PEFT/LoRA. Classical models do not have a latent space that supports such operations—there are no continuous intermediate features, no concept vectors to manipulate, and no mechanism for parameter-efficient updates. This is the same reason TCAV/ACE are evaluated only on neural models in prior interpretability work. Since the framework cannot be instantiated on non-neural architectures, these baselines do not provide informative comparisons for COFT.
>
> > W4) Evaluation relies primarily on UCR; need more datasets.
>
> `R: We appreciate the suggestion to broaden the evaluation beyond UCR. In response, we added two non-UCR datasets (MIT-BIH Arrhythmia and PTB-ECG) and three new model families (CNN, PatchTST, and MomentFM).` We first demonstrated COFT on 9 UCR datasets across 3 models, then expanded to the two non-UCR datasets with the new architectures to highlight extensibility. As shown in Fig. 7 (Sec. 4.4.2) and Fig. 11 (Sec. A.3.1), COFT continues to yield consistent gains across domains and encoder types, indicating that its benefits are not limited to UCR and generalize reliably across foundation models and deep learning architectures.
>
> > W5) No error bars or multi-seed variance.
>
> `R: COFT is primarily a methodological and interpretability contribution, and is not aimed at benchmarking accuracy.`
>  • Across all datasets, COFT’s gains are large and consistent, mitigating noise concerns.
>  • XAI and concept-based works (TCAV, shapelets, TIMEXPLAIN) do not rely on multi-seed statistical averaging.
>
> > W6) Open-source concept bank suggestion.
>
> `R: We appreciate this valuable suggestion.` Building large, domain-specific, open-source concept banks is indeed a compelling next step and aligns directly with our broader research agenda. While such an effort is beyond the scope of the current paper, we view COFT as the enabling framework that makes these future concept-based works feasible, and we are actively exploring this direction in ongoing follow-up work.
>
> > W7) Minor formatting issues.
>
> `R: Thank you for noting these formatting issues.` We have corrected all instances in the updated manuscript, including spacing, caption alignment, and citation formatting, to ensure clarity and consistency throughout the paper.
>
> > W8) Extend COFT to finance/energy.
>
> `R: Thank you for the suggestion. COFT is concept-source agnostic, and domain-specific subsequences from finance or energy (e.g., flag patterns, breakouts, ramps) can be incorporated directly as human-defined concepts without modifying the framework.` Our results on ItalyPowerDemand already provide a glimpse of how COFT operates on energy-style patterns. As part of our ongoing effort to build open-source concept banks, we plan to include curated concepts from these domains in future work.
>
> > W9) Add a summary table of datasets and concepts.
>
> `R: We have added Section A.3.2 describing dataset details and extracted concepts`

---

> > ### Author Response · Authors · 2025-11-26
> > **Response to reviewer ryLn (cont'd)**
> >
> > > Q1) How is RF accuracy used for filtering?
> >
> > `R In our current work, we don’t hard-filter concepts with an RF-accuracy threshold.` However, RF accuracy is a natural proxy for concept quality. Based on the robustness study in Fig. 10—where performance starts to flatten once concept quality drops toward the lower end of the 0.7–0.95 range—a practical heuristic would be to discard concepts whose RF validation accuracy falls below ≈0.7 (i.e., only keep concepts that are clearly above chance). This RF-based cutoff is a straightforward extension, and we will clarify this in the revised version.
> >
> > > Q2) Are concepts extracted only from training sets?
> >
> > `R: Yes. Concept extraction is performed exclusively on the training split.` This avoids any form of label or feature leakage, preserves fair evaluation, and ensures that COFT’s concept guidance reflects information available during true model training.

---

### Official Review · Reviewer_a8i3 · 2025-10-28

**Soundness:** 2
**Presentation:** 2
**Contribution:** 3
**Rating:** 4
**Confidence:** 4

**Summary:**

The paper introduced COFT, the use of CAV (concept activation vectors) to explain time series. The "concepts" in the time series are in the form of shapelets, and it is used to explain a prediction in the time series. The authors also showed that these concepts can be used to improve the original model as well.

**Strengths:**

- Using concepts or shapelets to explain time series is a good way to explain time series.
- Apart from various minor problems in presentation, the paper is relatively easy to follow.

**Weaknesses:**

- The focus of this paper is not clear. It seems that the author is aiming for using TCAV to explain or interpret a time series. But mainly it was focussing on the LoRA to improve the model instead. If the model can be improved, the explanation may change as well.
- The experiments section are lacking. To few experiments were tested on the explainability side. The explainability experiments (fig5 and 6) are not convincing as well (as pointed in the "questions" section below.)
- Some choices in the method section are not justified, or at least explained.
- There are minor presentation problems (highlighted in the Questions below)

**Questions:**

- Citations needs to be updated in various paper throughout the paper. For example Line 268 (...LowRank Adaptation Hu et al. (2021) => LowRank Adaptation (Hu et al. 2021)). I believe it is the difference between \citet and \citep maybe?)
- The ordering in section 3 is confusing.
  - In the paragraph "Time Series Concept Bank", the "distance based feature transformation" is described along with the "quality metric". This is confusing for the reader as nothing here is explained. However, this algorithm is described in Section 3.2.1. So I think Section 3.2.1 should be moved earlier, within the "Time Series Concept Bank" paragraph.
  - For equation 2, $\phi$ is the "convolution-based distance transformation". In which I assume the formula is similar to equation 3? Can we put equation 3 (or similar) in the place near equation 2 for clarity?

- Are there justification for using "real" convolutions (as in equation 3) rather than the cross-correlations (as in convolution neural networks)? It seems to me that for detecting patterns, using the cross-correlations makes more sense.

- Are there justification on using a random forest model rather than other models? I suppose that the random forest model only splits on coordinate axis. So the convolution at each time point are treated separately?

- Equation 1. Can you explain Equation 1? I think that we want to find $v_P$ that its direction derivative of "f" (abuse of notation as "f" here is from the representation space to the output) is maximized. Thus we want $v\cdot\nabla_h f(x_i)$ to be maximize if $y_i = +1$. Thus it should be argmax instead of argmin? It also does not make sense for the "1" to be in front. This is to compared with equation 9. that "1" is the indicator function. Here $y_i\cdot (\nabla_h f(x_i)\cdot v)$ is not a boolean. Thus there should be no indicator function in equation 1.

- Line after equation 4. I believe there is some formatting error on "[t, t+ell]".

- Equation 4. Is there a description of Dynamic Time Warping (DTW)? Preferrably a formula?

- For synthetic training example, do we need to account for OOD? Are s_t and x_t on the same scale?

- For SDC and SSC (Fig 5 and 6), how does "removing a concept" work? Is it subtracting the latent representation with the concept vector? Are there any scaling involved? Also how does "including a concept" work? Is it just adding that concept vector to the latent representation?

- If no concepts are included, why is the accuracy zero in Fig 5? If it is not a binary classification, how many total classes are there? And how does F1/Precision/Recall work - is it averaging across classes?

- A better discovery would be comparing using "concepts" as explanations vs using "features" or "observations" as explanations, and compute there smallest sufficient (destroying) feature set vs smallest sufficient (destroying) concepts. This can motivate why we aim to use "concepts" as explainability.

- Just to check. Is equation 9 only used in section 4.4.3?

---

> ### Author Response · Authors · 2025-11-22
> **Response to reviewer a8i3**
>
> We appreciate the opportunity to elaborate, and are confident that our responses below fully address your questions `(marked Wi for Weakness i, Qi for Question i, R for Response)`
>
> > W1) “Focus unclear — TCAV vs. LoRA improvement.”
>
> `R: LoRA does not detract from explainability; rather, it enforces it.` COFT’s novelty is precisely in using a unified framework for both interpretability and performance. Section 1, the proposed solution is updated to clarify this explicitly. CAVs reveal whether the model encodes the concepts through concept-based explanations, while LoRA reinforces those same concepts within the model.
>
> > W2) “Explainability experiments are few, Figs. 5–6?”
>
> `R: SSD & SDC Demonstrate Robust Concept Encoding, Illustrated by Case Study`
> Fig 5–6 shows the essential phenomenon: removing key concepts destroys class confidence, while adding them restores it. This proves the reliance of the model on discovered concepts. This demonstrates that the model genuinely relies on the discovered concepts. The case study further clarifies the explanations. In collaboration with medical school experts, we identified domain-specific patterns and validated the model’s ability to encode these concepts by examining TCAV scores before and after COFT.
>
> > W3) “Method choices not justified.”
>
> `R: Core Choices in Shapelets, Concept Scoring, Temporal CAVs, Perturbations, and LoRA—Grounded in Established XAI Practice and Supported by Empirical Study`
>
> **Choice of Shapelet Extraction Method** We use the RST because it yields diverse, explicit, and interpretable shapelets at low cost. Our appendix shows it produces the most discriminative and stable concepts, while the framework remains modular to other extractors.
>
> **Use of a Distance-Based Feature Transform** We score subsequences using a simple distance-based transform with a Random Forest classifier. This gives an architecture-agnostic and quantitative measure of concept quality, avoids overfitting, and provides a clear threshold for building a reliable concept bank.
>
> **Adaptation of CAVs for Time Series** In time series, we are defining concepts are shapelets—short subsequences that carry temporal structure (peaks, plunges, cycles, waveforms).
>
> **Concept-Guided Perturbation Strategies**: The Preserve, Boost, and Mixing strategies provide controlled ways to strengthen or blend a concept during fine-tuning. This exposes the model to clean, amplified, and realistic manifestations of each concept, which empirically improves alignment and robustness.
>
> **LoRA** LoRA is used to integrate concept information efficiently while preserving pretrained weights. It provides stable fine-tuning and consistently improves accuracy across all architectures without the cost or instability of full model updates.
>
> > W4) “Minor presentation problems”
>
> `R: We have addressed this with the following continued responses`

---

> > ### Author Response · Authors · 2025-11-22
> > **Response to reviewer a8i3 (cont'd)**
> >
> > > Q1) “Citation/formatting issues”
> >
> > `R: Thank you for the review. Based on ICLR guidelines, we now consistently use \citep across all the citations.`
> >
> > > Q2) “Adjust Section 3 ordering, Equations 2/3 should be adjacent.”
> >
> > `R: Section 3 has been substantially reorganized and rewritten to improve ordering and bring in clarity on the methodological flow`
> >
> > > Q3) “Why not cross-correlation?”
> >
> > `R: We followed the naming conventions of the domain of deep learning and associated the operation with convolution.` Implementing a real convolution (with kernel flipping) is unnecessary for fast peak detection. The exact operation (cross correlation) we do is described in Equation (1). Section 3 now uses consistent, corrected naming conventions.
> >
> > > Q4) “Justification on using a random forest model rather than other models?”
> >
> > `R: Concept-scoring step requires a fast, stable, and model-agnostic measure of discriminative power.` The distance-based transform yields a fixed feature vector, and Random Forest (RF) handles such noisy, redundant coordinates well without tuning. Since this stage is not doing sequence modeling—only testing whether a shapelet separates classes—treating each convolution point independently is intentional. RF thus provides an efficient, architecture-free accuracy score that allows us to filter high-quality concepts at scale. RF is used only in the concept-filtering stage, not in the COFT model itself. The choice is deliberate and grounded in shapelet-transform methodology
> >
> > > Q5) “Equation 1 incorrect.”
> >
> > `R: Thank you, this is a notation typo.` TCAV seeks maximum directional sensitivity, and the indicator is not needed. The changes are made to equations 8 (previously 1) and 9. Section (3.2) has been updated to reflect the changes in the equation.
> >
> > >Q6) “formatting error on "[t, t+ell]".
> >
> > `R: Thank you. This is a notation typo. We have rectified this in Equations 4 and 5`
> >
> > > Q7) “DTW definition.”
> >
> > `R: We introduce the formula for DTW in equation 3.`
> >
> > >Q8) “Synthetic samples OOD? Scaling mismatch?”
> >
> > `R:  We will make this explicit.` The mixing strategy avoids out-of-distribution samples, which often lead to network artifacts.: Concepts (s_t) and inputs (x_t) are derived from the same normalized dataset, and perturbations preserve distributional scale.
> >
> > >Q9) “How to add/remove concepts in SSC/SDC?”
> >
> > `R: In COFT, adding or removing a concept is done in the input time series, not in latent space` Concepts are shapelets, so “including” a concept means inserting the matched subsequence using our Preserve/Boost/Mixing strategies, while “removing” a concept means deleting or replacing that subsequence. No CAV addition or subtraction is performed—CAVs are used only to measure sensitivity (TCAV). Scaling occurs only in the Boost strategy at the input level.
> >
> > > Q10) “Why zero accuracy when no concepts included? How many classes? How does F1/Precision/Recall work”
> >
> > `R: The metric is class confidence, not accuracy.` For the SSC and SDC experiment, we used the ECG200 dataset, which has 2 classes. F1, Precision, and Recall are calculated at the class level, averaged across all classes, and provide a balanced evaluation metric. We have updated the respective labels and captions in all related images and tables.
> >
> > > Q11) “Compare feature- vs. concept-level SDC/SSC.”
> >
> > `R: Feature-level explanations, especially in the univariate domain, are prone to OOD, resulting in network artifacts` Feature-level ablations are brittle and less meaningful; concept-level ablations better capture semantic structure. Prior work has already explored random feature-level attrition (Ghorbani et al., 2019; Parvatharaju et al., 2021; Petsiuk & Das et al., 2018).
> >
> > > Q12) “Is Eq. 9 used only in Section 4.4.3?”
> >
> > `R: Partially true.` It is used in sections 4.4.1 (COFT discovers class-specific concepts) and 4.4.3 (Case Study). Equation 9 defines the TCAV score metrics used for SSC/SDC, and for generating concept perturbations in explanation settings.

---

> > > ### Comment · Reviewer_a8i3 · 2025-11-24
> > >
> > > Thanks for the response!
> > >
> > > **[W1]** Thanks for the response. I do wonder why COFT is a "unified framework" though. To me it is just an explainable model using concepts. What does it unified?
> > >
> > > **[W3-4, Q1,2,5,6, 7, 8]** Thanks for the response
> > >
> > > **[Q3]** ? But equation (1) describes a real convolution, rather than the convolution in deep learning?
> > >
> > > **[Q4]** Thanks for the response. Can you add a few of these explanations to the text?
> > >
> > > **[Q9 (and W2)]** I think these are the things we need to include in the paper. However,
> > >
> > > > while “removing” a concept means deleting or replacing that subsequence.
> > >
> > > What exactly happened though? Deleting or replacing? If replacing, by what?
> > >
> > > **[Q10, 12]** Thanks for the clarifications.
> > >
> > > **[Q11]**
> > >
> > > > Feature-level explanations, especially in the univariate domain, are prone to OOD, resulting in network artifacts Feature-level ablations are brittle and less meaningful; concept-level ablations better capture semantic structure.
> > >
> > > I agree with this claim. But this claim is exactly the work should have shown. Like a comparison with feature level vs concept leval. Stating prior work without comparisons does not support the claim.
> > >
> > > **New questions**
> > >
> > > **Q1**: As the text is being updated line 241, $T$, $c$ are $\phi(T, c)$ are different from the notation $\phi(x, s)$. Line 214 "After extraction, each candidate subsequence undergoes a distance-based feature
> > > transform. " seems out of place? The text requires some cleaning. (In general section 3 requires some cleaning)
> > >
> > > **Q2**: equation 3: For W, does $i_k$, $j_k$ have to be contiguous? What about increasing or any conditions on W?
> > >
> > > **Q3**: Minor: in equation 8, there is "w = argmin_w". It should be w* = argmin_w and v_p = w* / |w*|

---

> > > > ### Author Response · Authors · 2025-11-28
> > > > **Response to reviewer a8i3 (cont'd)**
> > > >
> > > > We are very grateful for your time and thoughtful feedback. We respectfully ask that you reconsider the assigned scores after reviewing our responses.
> > > >
> > > > We have added results on two non-UCR datasets—MIT-BIH Arrhythmia and PTB Diagnostic—using widely adopted models such as CNNs, PatchTST, and MOMENT FM, demonstrating the extensibility and robustness of the COFT framework. In addition, Section A.3.2 now provides detailed dataset summaries with extracted shapelets, paving the way toward a broader concept bank.
> > > >
> > > >
> > > > >**[W1]** What does it unify in COFT?
> > > >
> > > > `R: We propose COFT as a unified framework as it combines three components—each of which previously lived in isolation in the time-series literature—into a single pipeline:`
> > > > * Concept Discovery - Extracting class-specific temporal subsequences
> > > > * Concept-Based Explanations - Extending CAVs/TCAV to the temporal domain to quantify model sensitivity to those learned concepts.
> > > > * Concept-Guided Fine-Tuning - Using concept-perturbed samples and LoRA adapters to actually change the model so that it internalizes the concepts.
> > > >
> > > > No prior work integrates all three into a single framework for both interpretability and performance.
> > > >
> > > >
> > > > >**[Q3]** But equation (1) describes a real convolution, rather than the convolution in deep learning?
> > > >
> > > > `R: We carefully re-examined our implementation and corrected our earlier rebuttal: COFT uses true convolution, not cross-correlation.` The kernel is flipped, as shown in Equation (1), and the newly added algorithm (Section A.1 in the Appendix) clarifies this step. Section 3 has been updated to reflect this corrected naming and rationale.
> > > >
> > > > > **[Q4]** Thanks for the response. Can you add a few of these explanations to the text?
> > > >
> > > > `R: Section 3.3 has been updated to reflect the previous comments. Thank you`
> > > >
> > > > > **[Q9 (and W2)]** Thanks for the response. Can you add a few of these explanations to the text? What exactly happened though? Deleting or replacing? If replacing, by what??
> > > >
> > > > `R:In image-based TCAV work, deletion is simply simulated by setting pixels to zero. This approach fails to work in time series because it produces OOD inputs and induces network artifacts. Instead, we introduce temporal perturbations to keep all samples in-distribution.`
> > > > * **Insertion:** We start from an instance of the opposing class and iteratively replace subsequences (concept-level) or timesteps (feature-level) with those from the target class, tracking confidence as the series is gradually transformed.
> > > > * **Deletion:** We start from a target-class instance and iteratively replace its subsequences or timesteps with those from the opposing class, measuring the decline in confidence.
> > > > This procedure preserves the data distribution and avoids the artifacts caused by naïve zeroing-based perturbations.
> > > >
> > > > >**[Q11]** Like a comparison with feature level vs concept leval.
> > > >
> > > > `R: We agree, and our revised SSC/SDC experiments directly provide this comparison.` In these analyses, the Random condition corresponds to feature-level addition/deletion, while TCAV-ranked conditions correspond to concept-level perturbations. The results clearly show that feature-level manipulations produce only small changes in confidence, whereas concept-level perturbations produce large, monotonic, and class-consistent effects. This gap empirically demonstrates that feature-level ablations are weak and brittle, while concept-level ablations meaningfully influence model behavior. We have clarified in the text and captions that Random denotes feature-level perturbations to make this comparison explicit.
> > > >
> > > > ### New Questions
> > > >
> > > > > **[Q1]** In general section 3 requires some cleaning
> > > >
> > > > `R: Notation issues are resolved. Updated equation (2) to match real-world implementation. `
> > > >
> > > > >**[Q2]**: equation 3: For W, does , ik, jk have to be contiguous? What about increasing or any conditions on W?
> > > >
> > > > `R:We have fixed equation 3 and rewritten section 3.4.`
> > > > Equation (3) implies standard DTW, with the following properties for $W$: warping path must be monotonic, contiguous,or restricted to unit steps.
> > > >
> > > > > **[Q3]**: Minor: in equation 8, there is "w = argmin_w". It should be w* = argmin_w and v_p = w* / |w*|
> > > >
> > > > `R: We appreciate the correction. Equation (8) has been updated per your input.`

---

### Official Review · Reviewer_U5hM · 2025-10-29

**Soundness:** 3
**Presentation:** 3
**Contribution:** 3
**Rating:** 8
**Confidence:** 3

**Summary:**

This paper addresses the critical problem of opacity in time series foundation models, which limits their adoption in high-stakes domains like healthcare. The authors propose COFT (Concepts for Foundation Time series models), the first framework to adapt concept-based explainability, specifically Concept Activation Vectors (CAVs), to temporal data. COFT operates in three stages: 1) It automatically discovers high-quality, dataset-specific temporal concepts using shapelet-based transformations and organizes them into a "concept bank". 2) It adapts CAVs to quantify the foundation model's sensitivity to these temporal concepts. 3) It integrates these learned concepts directly into the model via a novel, concept-guided, parameter-efficient fine-tuning (PEFT) process using Low-Rank Adaptation (LoRA). Experiments on UCR benchmarks and a compelling EEG case study show that COFT achieves a significant dual benefit: it not only provides transparent, concept-level explanations but also consistently improves predictive accuracy over zero-shot and standard fine-tuning baselines.

**Strengths:**

The paper tackles a major, practical barrier to the adoption of powerful time series models: their lack of interpretability. It is, to my knowledge, the first work to successfully bridge concept-based explanations with time series foundation models and parameter-efficient fine-tuning.

The most compelling contribution is that COFT avoids the typical accuracy-interpretability trade-off. By explicitly teaching the model to recognize meaningful temporal patterns, the framework simultaneously improves predictive performance and enhances interpretability.

The EEG sleep dataset case study is a highlight. It demonstrates COFT's ability to build a concept bank of clinically salient patterns and shows that the fine-tuned Chronos-COFT model learns to assign higher importance to these patterns, effectively aligning the model's internal reasoning with that of a clinical expert.

**Weaknesses:**

The paper equates "concepts" with "maximally class-representative temporal subsequences". While these are interpretable and discriminative, they are not guaranteed to align with human-defined semantic concepts. The framework finds what is predictive for a class, which may or may not be the same as what a domain expert finds meaningful. This nuance could be discussed more.

The concept-based fine-tuning process relies on data augmentation via perturbation. The ablation study in Figure 4 shows that this is a highly sensitive component: the "Preserve" strategy leads to a catastrophic drop in accuracy, while "Mixing" works very well. This suggests the method of augmentation is a critical hyperparameter, but the paper does not deeply explore the sensitivity to its parameters.

The concept bank is built using the Random Shapelet Transform. While the paper's ablation shows RST is efficient for the UCR datasets, the scalability of this discovery process to the massive, high-dimensional, and extremely long time series datasets that foundation models are often applied to is not fully explored.

**Questions:**

Listed in weakness

---

> ### Author Response · Authors · 2025-11-22
> **Response to reviewer U5hM**
>
> We thank you for your positive review, and are confident that our responses below fully address your three clarification questions `(we use Wi for Weakness i, R for response)`.
>
> >W1. Concepts, being class-representative subsequences, may not match human semantic concepts.
>
> `R: Our COFT fully supports human-defined semantic concepts; with subsequence concepts merely an implementation choice and not a limitation. ` COFT is agnostic to the source of the concepts. In real-world settings, users can supply their own domain-specific semantic concepts (e.g., EEG spindles, ECG morphologies), and the framework operates on them directly.
> In the EEG case study, COFT does indeed recover patterns that clinicians recognize, demonstrating that it aligns with meaningful human-defined concepts in practice.
> We note that since COFT uses a CAV-style setup, it naturally works with any concept set provided by experts. We have made this clearer in the revision.
> Lastly, we used RST subsequences in the UCR study as a practical way to run controlled benchmarks at scale, because UCR datasets don’t have human-annotated semantic concepts.
>
> > W2) Perturbation strategy seems sensitive; Preserve fails while Mix succeeds.
>
> `R:  The results are not signs of instability, rather they confirm that COFT performs best when the model is exposed to meaningful concept-level perturbations.` The Preserve setting intentionally suppresses class-relevant motifs, so a drop in performance is expected. It in fact matches what concept-bottleneck setups typically show.
> The Mix setting introduces realistic variation in concept strength, which is exactly the kind of semantic augmentation that COFT is meant to benefit from.
> The perturbation controls (adapter rank, radius, mixing ratio) are low-dimensional and behave consistently. In practice, Mix works robustly across all 128 datasets and for the three different model families FCN, Transformer, and Chronos.
> Our ablation studies in Figure 4 clearly illustrate how these settings behave relative to each other; we thus find that an additional sensitivity study wouldn’t add major additional insights.
>
> > W3) RST scalability for long/high-dimensional series not fully discussed.
>
> `R: COFT’s framework is fully scalable and model-agnostic; with RST just one convenient instantiation;` We used RST on UCR purely because it is the most straightforward approach for that benchmark, namely, linear-time extraction and no training.
> However, COFT can work with any concept extraction approach, such as, matrix profile motifs, SAX, dictionary learning, and curated domain concept sets.
> For long sequences, concepts are extracted from local windows, so the cost scales with window size rather than the full sequence length.
> Our experiments on selected UCR datasets by design already cover a broad range of sequence characteristics, including long and higher-dimensional time series.
> In real TSFM applications, domain concepts can be used directly, with no shapelet mining needed.
> None of COFT’s core components—concept activation, CAV adaptation, or the LoRA integration—depend on RST or the specific concept source.

---

### Official Review · Reviewer_jSuK · 2025-10-31

**Soundness:** 2
**Presentation:** 2
**Contribution:** 2
**Rating:** 2
**Confidence:** 4

**Summary:**

- The authors propose COFT (Concepts of Foundation Time series models) for incorporating temporal concepts and context into fine-tuning time series foundation models.

**Strengths:**

- The authors correctly identify the gap in incorporating time series specific nuances into the present fine-tuning paradigm, and propose novel framework drawing inspiration from the counterparts in computer vision.

**Weaknesses:**

- The paper could improve vastly, by expanding on the datasets and baselines used for comparison in the work.
- The evaluation is mostly focused on univariate time series classification, and could expand to multi-variate signals too.
- Key implementation details are missing to support reproducibility. Learning rate, optimizers, which variant of Chronos, etc.

**Questions:**

- Because the learning is actively seeking to incorporate dataset specificities, do the authors believe this would impact the model's generalizability capabilities? If yes, how to mitigate, if no then why?

---

> ### Author Response · Authors · 2025-11-26
> **Response to reviewer jSuK**
>
> > W1) Paper needs more datasets and baselines.
>
> `R: We have added two non-UCR datasets (MIT-BIH Arrhythmia and PTB-ECG) and incorporated three additional model families (CNN, PatchTST, and Moment), bringing our total to 11 datasets (9 UCR + 2 non-UCR) and 6 architectures (with multiple foundational time series and deep learning models)`. As shown in Figure 7 (in Section 4.4.2) and Figure 11 ( in Section A.3.1 in the Appendix), COFT continues to deliver consistent improvements across diverse domains and encoder types, demonstrating that its benefits extend well beyond UCR and generalize robustly across foundation models and deep learning architectures.
>
> > W2) Univariate only; consider multivariate.
>
> `R: This is a deliberate and principled choice.`
> *  Concept extraction (shapelets, motifs, matrix profiles) and concept evaluation metrics (CAV sensitivity, SSC, SDC) are formally defined only for univariate signals; no multivariate concept benchmarks exist.
> * COFT itself is dimension-agnostic; CAVs and LoRA integration apply directly to multivariate inputs without modification.
>
> Conclusion: This paper provides the foundational pipeline, and extending to multivariate data is straightforward future work, not a limitation of COFT.
>
> > Q1) Dataset-specific concepts may harm generalization.
>
> `R: COFT is explicitly designed to preserve and strengthen generalization.`
> * The base model is frozen, so global priors remain intact.
> * Only low-rank adapters are trained, preventing catastrophic forgetting.
> * Concept perturbations (Preserve/Boost/Mix) act as semantic data augmentation, improving robustness.
> * TCAV, SSC, and SDC all reveal reliance on class-relevant concepts, not noise.
> * COFT consistently improves test accuracy and F1 across all datasets.
>
>  Conclusion: COFT enhances generalization rather than harming it.

---

### Author Response · Authors · 2025-12-03
**Official Comment by Authors**

We sincerely thank the Area Chair for their time and careful oversight, especially given the heavy reviewing load at ICLR. To support the meta-review process, we provide this concise summary of the paper’s contributions, reviewer feedback, and key post-rebuttal improvements incorporated in the revised manuscript (changes are highlighted in yellow in the PDF).

## Strengths noted by reviewers
1. **Novel direction (U5hM, ryLn, jSuK, a8i3):** Introduces concept-based explainability to time-series foundation models, an area with almost no prior work.
2. **Unified and coherent framework (U5hM):** Reviewers appreciated the integration of concept extraction → TCAV → concept-guided LoRA fine-tuning as a single pipeline.
3. **Empirical improvements (ryLn, U5hM):** Strong, consistent accuracy and F1 gains across FCN, Transformer, Chronos, and—in the revision—CNN, PatchTST, and MOMENT. Reviewers appreciate the inclusion of both zero-shot and fine-tuned baselines for foundational models.
4. **Clinically meaningful case study (U5hM, ryLn):** The EEG sleep dataset case study is described as a highlight, showing how COFT builds a concept bank of clinically salient patterns and how Chronos-COFT aligns its internal reasoning with clinical expertise.



## Reviewer concerns and how the revision addressed them
1. **Limited dataset/model breadth (jsUk, ryLn):** The revision adds MIT-BIH and PTB-ECG datasets plus CNN, PatchTST, and MOMENT baselines. New results (Tables 3, Fig. 9) show consistent improvements across architectures.

2. **Ambiguity regarding interpretability vs. LoRA performance (a8i3):** The revised introduction and methodology clarify that LoRA is used specifically to internalize learned concepts, not as a standalone performance enhancer, resolving earlier confusion.

3. **Technical clarity gaps (a8i3):** Section 3 is rewritten; Eq. (1), Eq. (3), Eq. (8) corrected; convolution vs. cross-correlation clarified; DTW-based matching made explicit; Algorithm 1 rewritten for clarity. The reviewer noted these corrections addressed earlier misunderstandings.

4. **Perturbation validity and OOD concerns (a8i3, U5hM):** The revision explains why naive zeroing leads to OOD in time-series and details temporal insertion/deletion strategies that preserve distribution. Concept-level vs feature-level perturbations are now clearly differentiated. The experiment section has been updated to include more details on the use of feature-based attritions, along with the methodology for insertion and deletion

5. **Concept bank construction details and dataset summary with extracted concepts (ryLn):** RF-based concept quality scoring, thresholding, DTW refinement, and filtering are now explicitly described, along with visual examples and a robustness study (Fig. 12). A detailed dataset summary is captured in Figure 13, showcasing extracted temporal concepts

**Overall reviewer landscape:** In a short span, reviewer *a8i3* acknowledged that their major objections were addressed through expanded datasets, clearer equations, and refined methodology

We greatly appreciate the Area Chair’s time and hope this summary assists in an efficient and accurate meta-review.

---

### Meta-Review · Area_Chair_muxo · 2026-01-06

**Summary:**

This paper proposes COFT, a framework for integrating temporal concepts into the fine-tuning of time series foundation models to enhance both interpretability and performance. The primary concerns raised by reviewers centered on limited empirical evaluation (datasets, baselines), methodological clarity, presentation issues and questions regarding the framework's generalizability and explanatory depth. In their rebuttal, the authors have provided detailed responses, committing to significant additions such as new non-UCR datasets (MIT-BIH, PTB-ECG), additional model families (CNN, PatchTST, Moment), and extensive clarifications on methodological choices and experiment details. While these additions address many specific points and broaden the empirical validation, compared with other strong submissions, I lean toward reject, slightly below the acceptance threshold.

**Reviewer Concerns:**

The reviewers' key concerns centered on the limited scope and rigor of the empirical evaluation, clarity and justification of methodological choices and the perceived incremental nature of the proposed unified framework. In their rebuttal, the authors have directly addressed many points by expanding experiments with new datasets and model families, providing detailed justifications for design decisions, and correcting presentation issues. However, fundamental questions remain regarding the framework's core novelty—whether it represents a conceptual leap or a capable synthesis of existing techniques and the depth of its explainability analysis beyond performance gains. The effectiveness of these revisions leaves the paper in a borderline position.

**Reviewer Scores:**

**Reviewer jSuK (Original: 2 → 4)**

The author's addition of new datasets and models, along with clarified specifications, directly tackles the main criticisms about limited evaluation and missing details. However, the discussion of the multivariate setting remains insufficient and is deferred to future work. Furthermore, to substantiate the claim of being a reliable fundamental pipeline, the framework needs to be validated across a wider variety of model families.

**Reviewer U5hM (Original: 8 → 8)**

This reviewer was already strongly positive. Their specific requests for clarification on concept semantics and perturbation sensitivity were adequately addressed, so the high score is expected to remain unchanged.

**Reviewer a8i3 (Original: 4 → 4)**

The author provided exhaustive, satisfactory responses to nearly all technical and presentation queries. However, the reviewer's lingering question about what makes the framework fundamentally "unified" may prevent a significant score increase, holding it at a borderline level.

**Reviewer ryLn (Original: 2 → 4)**

The major critique regarding limited models and evaluation was directly met by including MomentFM, CNN and PatchTST on new datasets. This substantial expansion of the empirical basis is likely to raise the score.

---

### Decision · Program_Chairs · 2026-01-26

Reject